# Discrete Event Modeling and Simulation for Reinforcement Learning System Design

**Laurent Capocchi** *,† and **Jean-François Santucci** †

SPE UMR CNRS 6134, University of Corsica, 20250 Corte, France; santucci@univ-corse.fr

* Correspondence: capocchi@univ-corse.fr
† These authors contributed equally to this work.

**Abstract:** Discrete event modeling and simulation and reinforcement learning are two frameworks suited for cyberphysical system design, which, when combined, can give powerful tools for system optimization or decision making process for example. This paper describes how discrete event modeling and simulation could be integrated into reinforcement learning concepts and tools in order to assist in the realization of reinforcement learning systems, more specially considering the temporal, hierarchical, and multi-agent aspects. An overview of these different improvements are given based on the implementation of the Q-Learning reinforcement learning algorithm in the framework of the Discrete Event system Specification (DEVS) and System Entity Structure (SES) formalisms.

**Keywords:** modeling; simulation; machine learning; reinforcement learning

## 1. Introduction

Artificial Intelligence (AI) concerns problems involved in Planing and Decision and has to control autonomous agents by defining a set of actions that an agent has to perform in order to reach a given goal from a known initial state. Machine Learning (ML) [1] is very often a good solution for these problems of Planing and Decision. The implementation of ML models and algorithms generally imply a lot of work because: (i) difficulties may appear to choose a model which will fit with the issue; (ii) for a chosen the model, best hyper-parameters are difficult to define; and (iii) classical ML tools are not able to associate the ML learning agents with a simulation framework due to the lack of temporal, multi-agent or hierarchical and dynamical aspects involved in the problem resolution. Associating Modeling and Simulation (M&S) with ML facilitates the resolution of problems such as: smart parking management or the management of healthcare systems. Furthermore simulation-based multi-agent ML [2] should help to solve the previous problems.

In this paper an approach based on the DEVS/SES (Discrete EVent system Specification/System Entity Structure) [3,4] has been proposed. The DEVS formalism has been proposed as a mathematical abstract formalism for the M&S of discrete event systems allowing an independence from the simulator using the notion of abstract simulator. SES allows to have to specify a family of DEVS models in terms of decomposition and coupling definitions.

This paper focuses on a set of new topics in the ML field and specifically in Markov Decision Processes (MDPs) [5] and Reinforcement Learning (RL) [6]. The following features have been proposed: (i) DEVS modeling RL feature based on agents and environment models; (ii) definition of RL temporal DEVS features; (iii) DEVS hierarchical modeling in RL system design; and (iv) DEVS-based multi-agents process. A description of a set of RL issues which are known as "difficult problems" and their resolution using DEVS and SES formalism has been proposed. In this approach RL systems designers have an way to analyze and efficiently resolve the previous problems.

The paper is organized as follows: Section 2 gives the background concerning MDP and DEVS formalism. The Section 3 describes how M&S is associated to ML concepts in

order to facilitate the design of ML environments. Section 4 points out: (i) DEVS explicitly separation between agent and environment involved in RL architecture; (ii) DEVS temporal aspects consideration in RL model behavior; (iii) DEVS hierarchy of abstraction in RL model development; and (iv) DEVS multi-agents in RL systems. Finally the last section of the paper gives some conclusions and proposes some perspectives.

## 2. Preliminaries

### 2.1. Markovian Decision-Making Process Modeling

Markov Decision Processes are specified as systems based on the Markov chains where future states depend only upon the present state [5,7]. A MDP is defined using the following set:

- S: corresponding to the state space.
- A: corresponding to the set of actions used to control the transitions.
- T: corresponding to the time space.
- r: corresponding to the reward function associated with state transitions.

The Bellman equation [8] is used to find a policy with no reference to a transition matrix involved in classical methods used in small-scale MDPs. The equation (Equation (1)) highlights that the reward considered in a long term time associated with an action is computed using the reward of the current action. The Q value associated with a state (state) and an action (action) must be computed as the addition of the reward (reward) and the expected future reward ($\gamma$) estimated when considering the next state (s'). Discount factor $\gamma$ allows to represent at long term ($\gamma = 1$) and even medium term ($\gamma < 1$) the next Q values at long term ($\gamma = 1$) and even medium term ($\gamma < 1$). Following the Temporal Difference [6] process, the Q matrix is computed according to the following equation:

$$Q(state, action) = Q(state, action) + \alpha[reward + \gamma(\max_{na'} Q(ns', na')) - Q(state, action)] \tag{1}$$

with $ns' \in S$, $na' \in A$, discount factor $\gamma \in [0,1]$ and learning rate $\alpha \in [0,1]$.

The Q matrix variable allows to compare future rewards towards the current reward. The best policy is computed using the Q-Learning process which is used to compute in a loop the best Q-value. Therefore this permits to choose the action $a$ whose $Q(state, action)$ is the maximum among from all the considered actions.

The Q-Learning process [9] is a well-known Reinforcement Learning algorithm used to resolve MDP problems. The Q-Learning pseudo-code [6] is given in Algorithm 1. In line 1, the initial state is defined before the repeat statement (line 2) allowing to try to obtain the final state (line 7). The Q matrix is updated at each step according to the Equation (1) by involving a new tuple (action, reward, s'). A new episode is considered each time the final state is computed.

---

**Algorithm 1** Q-Learning algorithm from [6].

---

1: Initialize Q(state,action)
2: Loop for each episode:
3:     Choose action from state using policy derived from Q
4:     New tuple (action,reward,s')
5:     Q <-Updated(Q)
6:     state <- s'
7: Until state is final state

---

From [10], the Q-learning algorithm converges according to the learning rates. But if $\gamma \simeq 1$, the value of Q can deviate [11].

In the case of the goal-reward process, the convergence of the Q-Learning algorithm may be assured by considering the Q matrix until a steady value is computed. "Repeat until Q converges" can be used in Line 2. However even if the Q-learning process is based on an efficient algorithm, the lack of generality should be highlighted. In order to offer

more generality to the process, DQN [12] may be used by involving neural networks. DQN uses a neural net to compute the Q-value function. The inputs to the neural net are actions, while the output expresses the Q value associated with the actions.

Although DQN has had a strong influence in great dimensional problems, the search space may remain low-key. In this case the Deep Deterministic Policy Gradients (DDPG) [13] off-policy algorithm may be used.

### 2.2. The Discrete Event System Specification

The DEVS formalism [3] has been introduced in order to offer a hierarchical modeling and simulation of systems. Every systems that whose inputs correspond to events over time and compute values on outputs corresponding to events over time is similar to DEVS models. DEVS offers a simulation process compatible with different computer configurations (single or multi-core processors).

DEVS offers two types of models: (i) atomic models allowing to define function transitions to specify the dynamics of sub-systems; and (ii) coupled models allowing to define the association of models in order to obtain a new model. This hierarchy notion involved in the DEVS formalism is not able to specify an abstraction hierarchy since DEVS models are specified at the same level of abstraction.

An basic DEVS model (atomic model) corresponds to a final state machine involving states and transition functions depending on event occurrence. An internal transition function ($\delta_{int}$) is used to compute the new state when no events occurs while an external transition function ($\delta_{ext}$) is used to change the state of the model depending on inputs and current state if an events occurs. A life time function ($t_a$) is used to manage the life time of a state. The output function called $\lambda$ allows to generate output messages. A DEVS abstract simulator is proposed to generate the behavior of a DEVS model.

### 3. Modeling and Simulation and Reinforcement Learning

M&S and AI are two complimentary fields of science. Indeed the AI may help the "simulationist" in the modeling phase of system of systems that are difficult or impossible to represent mathematically [14]. On the other hand, M&S may help AI models failed to deal with systems for lack of simple heuristics. In [15], the authors point out that: "AI components can be embedded into a simulation to provide learning or adaptive behavior. And, simulation can be used to evaluate the impact of introducing AI into a *real world system* such as supply chains or production processes".

Systems that already use AI have to associate AI in the models [16]. AI elements may be inserted into the simulation system. In [17], a learning algorithm is used to associate load balancing and bounded-window algorithms dedicated to gate level VLSI (Very large-scale integration) circuits simulation.

Figure 1 gives the potential associations of M&S features into ML processes. ML uses the classical three kinds of methods [6] in order to obtain models that are able assure the input sets to predict output variables. Figure 1 also shows that ML problems can be considered using Monte Carlo simulation based a "trial and error" process. Monte Carlo simulation is also able to generate random outputs using ML techniques. Optimization based on ML techniques is also an issue for integration ML into M&S. Agent-based models require both several hyper-parameters and important execution times in order to drive through all their combination to determine the best model structure. This design feature can be accelerated using ML. The configuration step of AI algorithms can also be accelerated using Simulation. The experimental replay generation can be also facilitated using simulation. RL elements may be also defined in order to represent rule-based models: for instance, in [18], the authors point out that an behavioral model is able to be generated (Output Analysis in Figure 1) by taking in to account a precise behavior defined by the inputs/outputs correspondences.

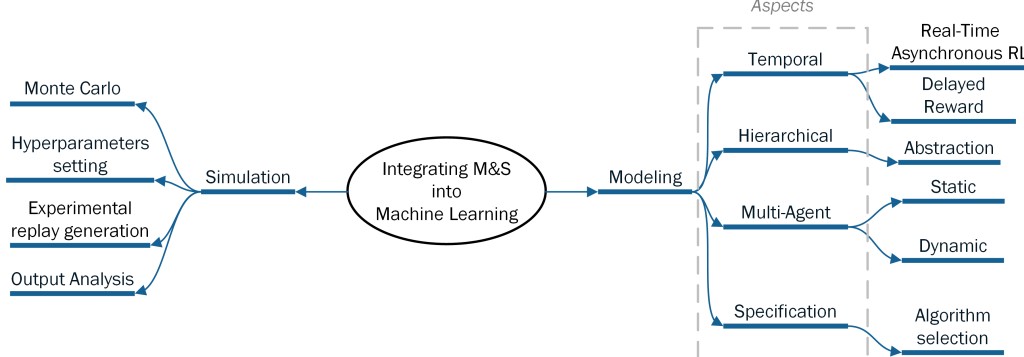

**Figure 1.** M&S aspects impacted when they are integrated into the machine learning algorithms.

Concerning the modeling features, the following items defined to facilitate ML deployment have to be pointed out:

- the temporal feature can be found in the RL processes. For example, MDPs provide both discrete and continuous time to represent the state life time [6]. In RL processes, the time notion in the rewards consideration is used to provide a way to model the system delayed response [19]. The concept of time involved in RL systems also offers to perform asynchronous simulations.
- the abstraction hierarchy makes possible the modeling of RL elements at several levels of abstraction. This allows to define optimal solutions associated with levels of details [20].
- multi-agent definition may be introduced in RL when the best possible policy is determined according to the links between dynamic environments involving a set of agents [21].
- grouping the ML elements required to bring solutions may also be difficult [6]. A set of ML techniques can be find to have to be considered and the selection of a particular algorithm may be hard. This selection can be facilitated by introducing models libraries.

This paper focuses on the interest of the DEVS formalism to help the RL system design with introducing temporal, hierarchical, and multi-agent features.

## 4. Discrete Event System Specification Formalism for Reinforcement Learning

DEVS is able to help the design of RL system as pointed out in Figure 2:

- The Data Analysis phase allows to select which kind of learning methods (supervised or unsupervised or reinforcement learning) is the most appropriate to solve a given problem. Furthermore, the definition of the state variables of the resulting model is performed during this Data Analysis phase which is therefore one of the most crucial. As pointed out in Figure 2, SES [22] may be involved in order to generate a set of RL algorithms (Q-Learning, SARSA, DQN, DDQN, ...) [6].
- DEVS formalism may facilitate the learning process associated with the previously defined model. Therefore DEVS is able to deal with the environment component as an atomic model. But the environment component associated to the learning agent as usual in RL approach may also be defined as a coupled model involving many atomic components. Therefore the environment may be considered as a multi-agent component modeled in DEVS whose agents may change during simulation.
- The real-time simulation phase involves both using the real clock time during simulation and exercising the model according to inputs. DEVS is a good solution to perform real time simulation of policies.

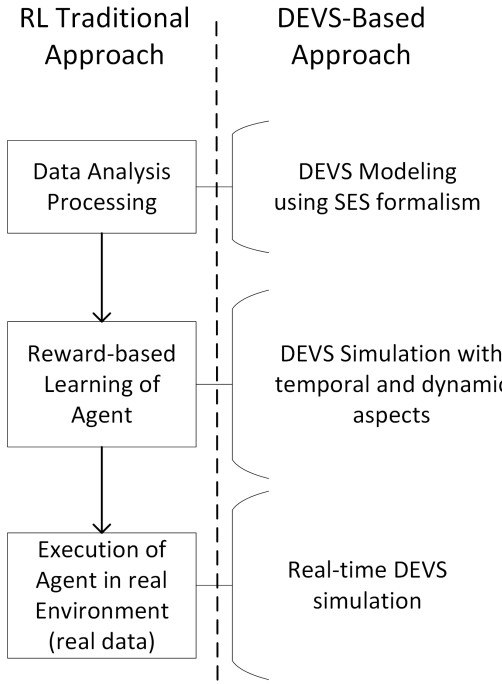

**Figure 2.** Traditional RL workflow to solve a decision making problem (RL Traditional Approach) and M&S equivalent tasks with the DEVS formalism (DEVS-Based Approach).

A RL model shown in (Figure 3) points out that the Agent and the Environment elements are easy to be modeled using DEVS atomic or coupled model according to the hierarchical adopted definition).

The association of DEVS and RL is achieved as follows:

- DEVS may use RL algorithms in two ways: when performing the modeling part of the system or when considering the simulation algorithms. DEVS is able to obtain benefits from an AI for instance, in order to help the SES modeling pruning phase or enhancing simulation performance using a neural net associated with the DEVS abstract simulator.
- AI processes may use the DEVS formalism in order to perform good selection of hyper-parameters for example. The presented focuses on this last problematic.

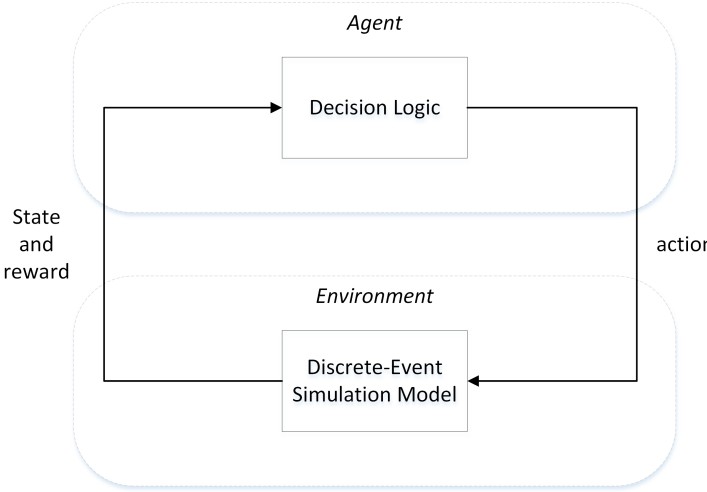

**Figure 3.** Learning by reinforcement.

The integration of DEVS into RL processes can be achieved as described below. The environment component and the agent element communicate to obtain to the best possible policy. Using the hierarchy notion inherent to DEVS, it is possible to improve the communication between agent and environment elements.

The Environment model answers to the messages sent by the agent model by defining a new tuple (*state*, *reward*) associated with an action that has been sent. Furthermore, the model indicates when the goal state has been reached using variable *d* (Boolean). The external transition function allows this previously introduces communication. The output function is then performed in order to convey the tuple (*state*, *reward*, *d*) towards the Agent element. State variables are updated using the internal transition function and no output is generated. The generation of episodes is performed by the model when a goal state is obtained.

Atomic model called Agent allows to answer to solicitations that come from the Environment component. If a tuple (*state*, *reward*, *d*) is received, the Agent element computes an action according to a policy associated with the chosen technique (*ε*-greedy). The model introduces a Q matrix modeled using a matrix used to define a selected learning process. Convergence allows to put the model into a state used to stop any communication with the environment component (STOP state). The Q matrix is updated using the external transition function according to the received triplet (*state*, *reward*, *d*). It is the internal transition function which is defined to allow the model to be put into the STOP state. The output function as usual is performed after the external function.

This section details how a RL system can be specified using DEVS using the temporal, abstraction hierarchy, and multi-agent features.

### 4.1. Temporal Aspect

When dealing with temporal aspects associated with RL methods, two features can be pointed out [6,23]:

- Implicit time aspect is indeed present in the Q-learning or SARSA algorithm since it leans on the notion of reward shifted in time. However, this implicit notion of time does not refer to time unit but only to time steps involved in the algorithms. The idea is to add a third dimension in addition to the classical ones (states and actions). In this case, the time feature is introduced even if the results give an ordering of the actions involved in the chosen policy.
- The explicit time aspect is not involved in the classical Q-learning and SARSA algorithms. However, a set of work introduces the possibility to associated random continuous duration to actions [24]. Time is made explicit in these approaches.

#### 4.1.1. Implicit Time in Q-Learning

The time aspect is introduced by adding a third dimension to the classical Q-learning equation. In this case, the Q-value function depends on the state, the action, and the stage $t$. This new dimension allows one to propose an ordering of the actions involved in the policy since the resulting best policy depends not only on the state and the action but also on the stage.

Figure 4 depicts the concept of the simulation analysis of the action-value function $Q$ during the evolution of the indexes. $Q_0$ is the matrix involved in the first simulation that includes the first index values (at simulation time 0). It consists of state/action tuples according to Bellman equation definition, and its size is the number of possible states on the X-Axis and the number of possible actions on the Y-Axis. One point $v_0$ on $Q_0$ (state/action in Figure 4) can be followed up during the evolution of the indexes from 0 to L. This path $(v_0,...,v_k,...,v_L)$ represents the evolution of the importance of the action $a$ for a corresponding state $s$. Performing a maximization on this path returns to find the optimal action-value function $Q_*$ or a best possible time to take the action $a$ for the state $s$:

$$Q_*(s, a, t) = \max_t Q(s, a, t) \qquad (2)$$

However, the size of $Q$ can change during the simulation due to the number of possible states that depend on the values of the indexes. Accordingly, it may be possible that a tuple (state/action) is not present in a $Q_t$ matrix during the simulation. In this case, this specific value (for a specific simulation time) is not considered for the time analysis in the Equation (2).

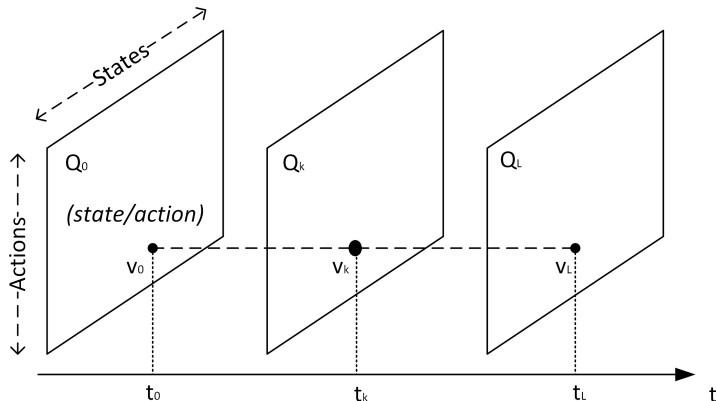

**Figure 4.** Q matrix simulation analysis during the evolution of the indexes.

4.1.2. Explicit Time in Q-Learning

In MDPs, the time spent in any transition is the same. This assumption, however, is not the case for many real-world problems.

In [25], the authors extend classical RL algorithms developed for MDP and for semi-Markov decision processes. Semi-MDPs (SMDPs) extend MDPs by allowing transitions to have different duration ($t$). In [23], the authors considered the problem of learning optimal policies in time-limited and time-unlimited domains using time-limited interactions (limited number of step $k$) between the agent and the environment models. Time notion is explicitly considered but only in terms of limited time $T$ considered for maximize the total reward assigned to the agent that try to maximize the discounted sum of future rewards:

$$G_{t:T} = \sum_{k=1}^{T-t} \gamma^{k-1} R_{t+k}$$

In [26], the authors introduce a new model-free RL algorithm to solve SMDPs problems under the average-reward model. In addition, in [27], the authors introduce the theory of options to bridge the gap between MDPs and SMDPs. In SMDPs, temporally extended actions or state transitions are considered as indivisible units; therefore, there is no way to examine or improve the structures inside the extended actions or state transitions. The options theory introduces temporally extended actions or state transitions, so called options, as temporal abstractions of an underlying MDP. It allows to represent components at multiple levels of temporal abstractions and the possibility of modifying options and changing the course of temporally extended actions or state transitions. In [28], the authors explore several approaches to model a continuous dependence on time in the framework of MDP, leading to the definition of Temporal Markov Decision Problems. They then propose a formalism called Generalized Semi-MDP (GSMDP) in order to deal with an explicit event modeling approach. They establish a link between the Discrete Event Systems Specification (DEVS) theory and the GSMDP formalism, thus allowing the definition of coherent simulators.

In the case of explicit time, the time of a transition has to be taken into account. The time between actions is an explicit variable (which may be stochastic) and dependent of the state and the action. This transition time is known as the sojourn notion. DEVS Markov models [29] are able to explicitly separate probabilities specified on transitions as well as defined on the times/rates. Furthermore the dynamic properties involved in the DEVS

formalism allows to dynamic modify these specifications during the model simulation. The transition probabilities are classically associated with a Markov chain while probabilities on transition times/rates are associated with the sojourn notion. This modeling features associated with Markov chains offers possibility to define explicitly and independently transition probabilities and transition times.

*4.2. Hierarchical Aspect*

Hierarchical aspect deals with top-down or divide and conquer approach: A MDP can be decomposed into a smaller MDPs organized in a hierarchical way. There are two benefits that can be obtained from the specification of a hierarchy in a MDP: (i) reduce difficulties of solving a problem by approaching it in the form of a set of sub-problems that are easier to solve individually. As a result both the state space and action space must be smaller, and thus the resolution may be facilitated; and (ii) specify actions at a higher level that may be reused according to different types of goals.

Thanks to the hierarchy concept, a hard task may be split into more simplest tasks. In [6] the basic tasks are specified as options that are implemented by a policy performed on part of the set of states. In [30], sub-tasks are built to solve the corresponding sub-goals which have been initially defined. Rewards related to the sub-goals are used to learn the sub-tasks. The author Dietterich pointed out that the definition of a MDP using a hierarchy of small MDPs is equivalent to the definition of the value function of the considered MDP as a composition of the value functions of the previously mentioned small MDPs.

In [31] a feudal system involving agents at different levels is proposed in order to implement the hierarchy in a learning agents. An agent situated at a given level obtains solicitations from an upper-level agent and sends solicitations to agents at a lower level. Only the agents specified at low levels are able to deal with the environment component.

In [32], Hierarchy specified in the environment is presented and the the concept of hierarchical Abstract Machines (HAM) named "partial policies" is defined. For instance, Parr's maze is designed using orthogonal hallways involving obstacles. In this case HAM specified at a high level deal with the selection of a hallway, intermediate HAM deal with the traversing of the hallway and lower-level HAM have to take into account avoiding obstacles. As explain previously, the concept of the hierarchy may decrease the state space involved in the RL process by learning a sub-task from only the relevant properties. The partial description of the environment is used by defining the HAM that allows to obtain the next state. A state space partition composed of agents related to only one part of the space is used to define the feudal agents. As stated before, the MDP resolution is based on the state space reduction but the fact to consider only a part of the context implied in the execution of the policy allow also to share and reuse the process more easily.

An example of feudal hierarchy is given in Figure 5.

The environment may be in a state belonging to *{S0, S1, ..., S11}*. The initial agent level is represented by *{A, B, C, D}* agents. Every agent has to take into account a reduced number of states. The following level is represented by {X, Y} agents and the highest level is represented by the {SUP} agent.

Using DEVS to implements a generic feudal agent points out the property of Feudal MDP [33] expressed as follows: since an agent specified at a low level of description relies on the same model than an agent defined at an upper level of description (this agent is used to control the agents defined at lower level), the set of agents defined a low level of description can be considered as the environment states. Therefore, the hierarchy defined on agents and on the environment is the same: the agents defined at a given level (let us called L) correspond to the environment when dealing with agents defined at the L+1 level. For instance, according to the n Figure 5, X the agent has to communicate with the environment whose state may be A and B.

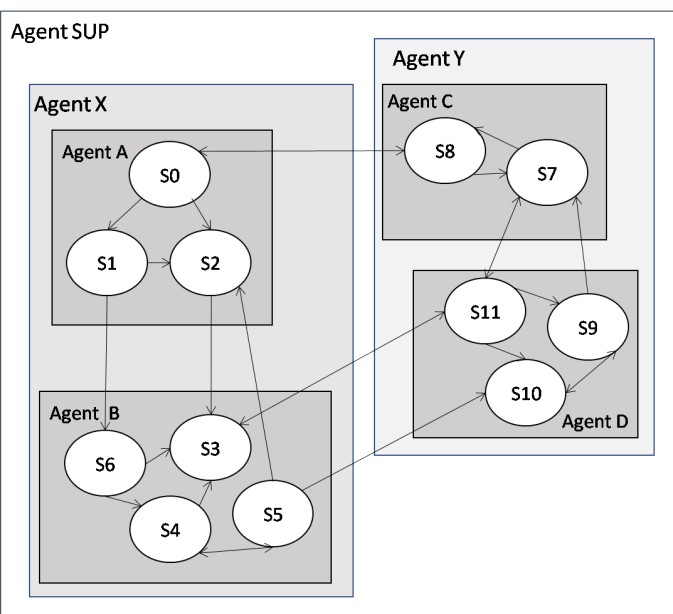

**Figure 5.** States aggregation mechanism to implement hierarchy in feudal agents.

### 4.3. Multi-Agent Aspect

This section briefly describes how DEVS/SES multi-agents may be introduced in RL.

The Markov decision process generalization of the towards multi-agent notion may be defined as a tuple $\{S, A_1, ..., A_m, p, r_1, ..., r_n\}$ where:

- m: the agents number;
- S: environment states;
- $A_i$, $i \in 1, ..., m$: actions associated to the agents and generating the action set $A = A_1 \cup A_2 \cup ...A_n$;
- $p : S \times A \times S \rightarrow [0, 1]$: transition probability function;
- $r_i : S \times U \times S \rightarrow R$, $i = 1, ..., m$: reward functions associated with the agents.

The state transitions are the result of the global action of all the agents. The policies $f_i : S \times A_i \rightarrow [0, 1]$ define jointly the policy $f$. Since the rewards $r_i^{k+1}$ lean on the global action, their results lean on the policy $f$. The Q-function associated with the agents leans on the global action and on the policy $f$. The complexity is great since it should be taken i to account that the state-action space exponentially depends on the number of states and actions. Because RL algorithms define values associated with all state or state-action doublet, this causes an exponential computational complexity expansion. This results in the fact that the complexity is more important in multi-agent RL than in single-agent RL.

In addition, when dealing with RL, many families of algorithms have been defined in [34]. Each algorithm presented in this paper points out a set of advantages but also a set of drawbacks and can be used depending on the context of the application.

Finally when proposing a Multi-Agents RL System, several types of architectures have to be considered. To give to the designer of a multi-agent reinforcement learning project the possibility of choosing between the previously mentioned architectures, a generic framework based on SES/DEVS formalism can be proposed. SES [22] is an ontology framework allowing to represent the elements of a system and their interactions using a family of DEVS models. A SES defines a set of hierarchical DEVS models. Each resulting DEVS model may be obtained by a pruning process which allows to choose an element of the set of considered models. The idea is to give a designer a library of agent models allowing to deal with the basic agent models (critic-only, actor-only and actor-critic) as well as to specify a set of multi-agents architectures using SES in order to define a family of simulation models. The pruning of the SES allows the designer to choose one particular multi-agent architecture and to execute simulations and learning.

## 5. Related Work

Machine learning algorithms have been used in the field of simulation in many cases [17,35–38]. For instance, in [35], the authors derive rules of distraction driving using Support Vector Machine based on data from a driving simulation environment. In [17], the Q-learning has been used to associate load balancing and bounded-window algorithms dedicated to gate level VLSI circuits simulation. A set of AI ML techniques have been involved in DEVS M&S. In [39] the authors deal with DEVS execution acceleration based on machine learning thanks to the explicit separation between modeling and simulation. In [40], DEVS simulation is used to help to set values associated with of hyper-parameters involved in neural nets. In [29], DEVS Markov features are presented as well as their implementation in MS4Me framework [22]. DEVS Markov involves the basic and enhanced features of Markov chains since DEVS modeling aspects are fully compatible with Markov concepts. In [28], a RL technique involving temporal features is defined using simulation in order to deal with generalized semi-Markov decision processes.

Concerning the improvement of the DEVS formalism based on machine learning algorithms , in [39,41], the authors add predictive machine learning algorithms in the DEVS simulator to improve simulation execution due to the separation between modeling and simulation inherent in the DEVS formalism. Furthermore a predictive model that learns from past simulations has been defined. In [42], the use of discrete event simulation with a deep learning resource has been defined in order to propose intelligent decision making in the form of smart processes. An another example combining AI and simulation can be found in [43] where a new machine-learning based simulation-optimization integration method has been proposed. The machine learning model allows to accelerate simulation-optimization integration to find optimal building plans.

When simulation and AI are combined, AI simulation models have to be accurate. For instance, in [44], a validation methodology for AI simulation models has been defined. The combination may also have nice impacts on specific applications [45–47]. In [45], the authors deal with a framework that incorporates Simulation Modeling and ML for the purpose of defining pathways and evaluating the return on investment of implementation. In the field of parallel simulation, a cache-aware algorithm that relies on machine learning has been defined in [46] in order to maximize data reuse, allow workload balance among parallel threads.

In [48], a machine learning algorithm has been involved to solve problems coming from the performance evaluation of simulations in the field of effective design space exploration. A methodology to obtain cross-platform performance/power predictions has been defined in order to derive the simulation performance/power on another platform.

In [38], a conceptual framework to help the integration of simulation models with ML has been defined.

All this work mainly consider the problems raised from the association of M&S and ML. None of the previously quoted work leans on temporal, hierarchical and multi-agent notions involved in the RL system design using DEVS. As described in [29], the ML (and more specially MDP) concepts of stochastic modeling are implicitly involved in discrete event simulation. Concerning the temporal aspect, DEVS allows to explicitly and separately deal with probabilities defined on state transitions and with probabilities of transition times. This advantage allows the designer to model temporal aspect in ML models and execute it in a simulation framework. Dealing with hierarchical aspect, integrating ML modeling into DEVS offers a wide variety of model types that may be hierarchically inserted within the same framework. DEVS allows to organize such models into classes that are associated with both the traditional ones encountered in the mathematics literature as well as the structural notions that define all DEVS models as specifications of input/output dynamic systems. Furthermore classes and sub-classes of such models that allows definition of multi-agent models may be formed by choosing elements belonging to some specializations that define a specific model architecture. DEVS models are able to represent complex ML systems at the level of individual subsystems and actors. Each system or actor may be defined

as a component with states and transitions as well as inputs and outputs that allow the interaction with atomic models belonging to coupled models.

## 6. Conclusions

This paper points out how DEVS features can be used in helping RL system design putting emphasis on the temporal, hierarchical, and multi-agent aspects. A panorama of a set of simulation-based RL system design features is given and four of them realized in the framework of DEVS are detailed: (i) DEVS modeling of the agent and environment RL components based on the hierarchy feature inherent in DEVS; (ii) DEVS modeling of RL temporal features since DEVS allows to improve both implicit and explicit time aspects definition in MDPs; (iii) DEVS hierarchical modeling in RL system design since DEVS allows to determine optimal policies based on abstraction level; and (iv) possibility to take into account RL multi-agents since DEVS/SES allows to choose one particular multi-agent architecture and execute simulations and learning. The future work can be described as follows: (i) to work on dynamic aspects of RL since DEVS involves the modification of models during simulation; (ii) to work on a hierarchy of RL agents based on DEVS abstraction hierarchy feature (initiated in [33]); and (iii) to work on the scalability of the RL proposed approach.

**Author Contributions:** All authors have contributed equally to this work. All authors have read and agreed to the published version of the manuscript.

**Funding:** This research received no external funding.

**Informed Consent Statement:** Informed consent was obtained from all subjects involved in the study.

**Data Availability Statement:** Not applicable.

**Conflicts of Interest:** The authors declare no conflict of interest.

## Abbreviations

The following abbreviations are used in this manuscript:

| | |
|---|---|
| M&S | Modeling and Simulation |
| AI | Artificial Intelligence |
| RL | Reinforcement Learning |
| ML | Machine Learning |
| MDP | Markov Decision Process |
| DEVS | Discrete Event system Specification |
| SES | System Entity Structure |

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
