# Peer review of "Discrete Event Modeling and Simulation for Reinforcement Learning System Design"

_information, doi:10.3390/info13030121_

Round 1

Reviewer 1 Report

The authors propose an approach based on discrete event Modeling and Simulation (M&S) using the DEVS (Discrete EVent system specification) and SES (System Entity Structure) formalisms for simulation-based multi-agent ML.

Apparently, this is the objective of the work, because by the organization of chapter 1 this is not very clear. I am used to seeing the presentation of the general context of the work, some aspects and/or relevant theories (even superficially), the gaps that deserve a research effort, the objective, and the general organization of the work.

Another point that catches my attention is that the objective is not consistent with the title (“Combining Discrete Event Modeling and Simulation with Reinforcement Learning: An Overview”) and abstract (“This paper details how discrete event Modeling and Simulation could be integrated into Machine Learning concepts and tools in order to improve the design and use of Machine Learning frameworks.”). From the title and abstract, my perception was of reading a survey on the topic, something exploratory, but reflective, which is different from making a proposal, where I would expect to see a comparison of the results obtained by other related works.

SES and multi-agent systems are dealt with in the text but not covered in the title or abstract. In other words, the general scenario of the work is not clear in the abstract. The abstract is also very superficial, not showing the results achieved. It is difficult for the reader to assess whether the work is relevant to him.

In Chapter 2.2, in "Discrete Event System", a doubt arose: what would be "The life time of a state is determined by a time advance function called ta"? In my understanding of DES, this type of system is event driven, and therefore time is irrelevant. It is a different approach to a classic control system. And it is also different from discretizing time, which is associated with sampling signals in continuous systems.

In Chapter 3 you quote "discrete and continuous time". This reinforces my understanding that you are mixing the concepts of "sampling at discrete intervals" with "discrete events".

The last step in Figure 3 is "Real time DEVS simulation". Here you are either talking about (1) "emulation" because it is in real time as opposed to simulation, which time can be accelerated, for example, or (2) you are talking about using real data. Most likely you are adopting understanding (2) based on the text starting at line 243 ("submit the model to the actual input data (test phase")).

Between lines 355 and 362, the following text is duplicated: "allows us to work with the times of the transitions. These can be referred to variously and equivalently as sojourn times, transition times, time advances, elapsed times, or residence times depending on on the context."

In Chapter 4.2, rather than the trip planning example, I think it would be better to explain that the "hierarchical aspect" is something similar to a top-down or divide and conquer approach.

For me Chapter 4.3 is confused, lost at work. How "This section mainly focus on the possibility to introduce multi-agents in RL" is related to the title and the abstract.

Chapter 5 looks good. It is related to the title. However, it is out of place and, as the authors present a proposal in the work, this proposal is not compared with these other works. They were just quoted.

Chapter 6 has to be revised based on the work you actually want to present. My final impression is that this is a large work, that covers many theories / concepts, and you wanted to focus on just a few aspects for this publication. However, you were not able to delimit the scope in order to have a clear work for the reader.

At least reference 44 must be corrected. Other than that, only 22 out of 48 references can be considered up to date. Many are 20 years old or more!

Some less relevant but important points:

  • Use "Figure" instead of "figure";
  • I believe "equation" is more suitable than "formula".

Reviewer 2 Report

This paper reviews the literature and examines the benefits of combining the algorithms of the Machine Learning framework with the Discrete Event Modeling and Simulation framework.

The structure of the paper is appropriate and the language is clear and easy to follow. Its English is good, with no spelling or typing errors.

The paper processes both classical sources on the subject and recent literature. There are only a few figures, but what is there is good. The usefulness of the "5. Related Work" section deserves special mention.

The paper could be more practical, but overall it is of good quality, I did not find any substantive deficiencies and I recommend its publication without changes.

Reviewer 3 Report

This paper discusses and explores the possibility of combining ML and discrete-event simulation.
Even though the topic is relevant and interesting, I personally find the paper very descriptive, with a large part of the text dedicated to a review of the literature (which is expected as the paper is presented as an overview).
I would however have expected more quantitative results, in the form of algorithms, codes, or concrete examples. Here the discussion remains quite abstract.
Besides and as minor comments, the abstract is not well formulated and should be rephrased, and the introduction is also not very clear. There are quite a number of grammatical mistakes and English should be improved.

Round 2

Reviewer 1 Report

The text got better. The work is clearer.
My observations:
- put ";" between items (i)...; (ii)...; and (iii)... for example;
- line 118: "...issues, (like..." >>"...issues (like...";
- line 122: "... degree of free random system is 10." - is it "... degrees of freedom of a random system"?
- item 2.2: is the entire text of this item based on reference 3?
- line 192: "In return, ..." >> ""In turn, ..."??
- line 203 to 225: what are the references?
- line 266 to 304: what are the references?
- line 478: "For example, In [32]..." >> "For example, in [32]...";
- line 521: "interesting" is a subjective word that is advisable to avoid;
- line 521 to 523: the text is better; however, the reader ends the paragraph with a doubt as to why all these characteristics together are important. The reader finds part of the answers in chapter 6, but I think there is still a better discussion, perhaps bringing contributions from chapters 3 and 4 to support the comparisons.

Reviewer 3 Report

The authors have indeed clarified the objectives of the article.

There are still quite a number of grammatical errors distributed all over the text, which require attention. The text should be double-checked carefully.

For instance (just a few, but there are more):

  • page 2: section xx gives/describes/etc. (no future tense)
  • page 3: the Q-learning algorithm converges
  • Page 5: Figure 2 depicts
  • Page 6: This paper focuses
  • Page 6: The DEVs formalism allows formally modeling
  • etc.
